# Murine Placental Erythroid Cells Are Mainly Represented by CD45^+^ Immunosuppressive Erythroid Cells and Secrete CXCL1, CCL2, CCL3 and CCL4 Chemokines

**DOI:** 10.3390/ijms24098130

**Published:** 2023-05-01

**Authors:** Kirill Nazarov, Roman Perik-Zavodskii, Olga Perik-Zavodskaia, Saleh Alrhmoun, Marina Volynets, Julia Shevchenko, Sergey Sennikov

**Affiliations:** Laboratory of Molecular Immunology, Federal State Budgetary Scientific Institution, Research Institute of Fundamental and Clinical Immunology, Novosibirsk 630099, Russia; kirill.lacrimator@mail.ru (K.N.); zavodskii.1448@gmail.com (R.P.-Z.); perik.zavodskaia@gmail.com (O.P.-Z.); saleh.alrhmoun1@gmail.com (S.A.); mrsmarinavolynets@gmail.com (M.V.); shevcen@ngs.ru (J.S.)

**Keywords:** erythroid cells, erythroblasts, CD71^+^ erythroid cells, CECs, placenta, allogeneic pregnancy, syngeneic pregnancy

## Abstract

Erythroid cells are emerging players in immunological regulation that have recently been shown to play a crucial role in fetomaternal tolerance in mice. In this work, we set ourselves the goal of discovering additional information about the molecular mechanisms of this process. We used flow cytometry to study placental erythroid cells’ composition and BioPlex for the secretome profiling of 23 cytokines at E12.5 and E19.5 in both allogeneic and syngeneic pregnancies. We found that (1) placental erythroid cells are mainly represented by CD45^+^ erythroid cells; (2) the secretomes of CD71^+^ placental erythroid cells differ from the ones in syngeneic pregnancy; (3) CCL2, CCL3, CCL4 and CXCL1 chemokines were secreted on each day of embryonic development and in both types of pregnancy studied. We believe that these chemokines lure placental immune cells towards erythroid cells so that erythroid cells can induce anergy in those immune cells via cell-bound ligands such as PD-L1, enzymes such as ARG1, and secreted factors such as TGFβ-1.

## 1. Introduction

Pregnancy is the process of embryo development in the uterus with the help of the placenta [1]. Pregnancy can be divided into allogeneic pregnancy (AP) and syngeneic pregnancy (SP). AP happens when there are two genetically distinct parents [2]. This is a usual scenario for humans and for genetically distinct mice. If parents are genetically the same, i.e., mice from the same line, the pregnancy is then called syngeneic pregnancy (SP) [3].

The placenta is a highly specialized organ that is formed during fetal development. The placenta’s primary function is to supply the fetus with nutrition and oxygen. The placenta is formed by two distinct cell types: the cytotrophoblast and the syncytiotrophoblast [4]. Some cytotrophoblasts invade the endometrium and spiral arteries during pregnancy to become extravillous cytotrophoblasts (EVTs) so that the ruptured spiral arteries can supply the placenta with blood full of nutrients and red blood cells [5]. Among these red blood cells are erythroid cells [5,6].

Erythroid cells are nucleated red blood cells that express the surface protein GPA (CD235a) and Ter-119 in humans and mice, respectively [7]. However, some researchers tend to use another marker for erythroid cell studies, namely CD71. Such cells are named CD71^+^ erythroid cells (CECs) [8].

Previously, it has been shown erythroid cells can produce cytokines [9,10,11]. This suggests that erythroid cells have immunoregulatory potential. Erythroid cells have a local immunosuppressive effect in the placenta, which is necessary for the mother’s body to accept the implantation of the embryo—to maintain fetomaternal tolerance. It was shown that the depletion of erythroid cells in pregnant mice led to an impairment of fetomaternal tolerance and activation of the maternal immune response against the embryo. Nevertheless, a decrease in the number of erythroid cells led to an increase in the recruitment of CD4^+^ and CD8^+^ T cells to the placenta, which was accompanied by an increase in the production of pro-inflammatory cytokines, including TNFα and IL-6. Moreover, the depletion of erythroid cells in pregnant mice reduced the concentrations of the anti-inflammatory cytokines IL-10 and IL-4, which play a central role in the development of the fetus and maternal tolerance. All of these effects were observed only in AP and resulted in complete fetal resorption, suggesting that erythroid cells play a key role in the development and maintenance of fetomaternal tolerance [8,12].

Human placental erythroid cells have significant immunomodulatory properties and strongly inhibit the proliferation of CD4^+^ and CD8^+^ T cells in vitro [13,14]. A significant proportion (up to 30%) of fetal erythroid cells can be found in maternal blood [15]. This may indicate that erythroid cells are involved in the formation of tolerance, given their pronounced immunosuppressive properties and the ability to produce immunoregulatory mediators, such as TGFβ-1 [16]. It was found that erythroid cells of umbilical cord blood more effectively suppress the inflammatory response of stimulated adult peripheral blood cells than umbilical cord blood monocytes, which suggests that fetal erythroid cells can suppress the maternal immune response and prevent inflammation and development of an unwanted immune response against the fetus [17].

During pregnancy, mild anemia is physiological; this is caused by an increase in plasma volume, which cannot be quickly compensated by a proportional increase in red blood cells. Elevated concentrations of estradiol and 27-hydroxycholesterol, acting through the estrogen receptor alpha ERα, promote hematopoietic stem cell division and, together with erythropoietin, promote erythropoiesis and an increase in the content of erythroid cells in the mother’s spleen, as well as an increase in the total mass of the mother’s spleen [18,19]. Erythroid cells in the spleen of pregnant mice have increased expression of EpoR and reduced expression of the death receptor Fas, which is associated with their increased proliferation and reduced apoptosis [20].

In addition to erythroid cells, Tregs, macrophages, and dendritic cells are involved in maintaining immunological tolerance during pregnancy. A special role is played by decidual Tregs with the CD4^+^CD25^+^FoxP3^+^ phenotype expressing CTLA4 and PD-L1 and secreting IL-10 and TGF-β, through which they maintain a local immunosuppressive state: the anti-inflammatory and tolerogenic phenotype of M2 macrophages and tolerogenic dendritic cells [21,22]. In the placenta, next to T cells, there are Bregs, which are characterized by the secretion of IL-10 and IL-35. Through these mediators, Bregs contribute to the induction and proliferation of Treg cells and also suppress the differentiation of Th17 cells [23,24].

The PD-1/PD-L1 pathway plays a role in the development and maintenance of fetomaternal tolerance. High expression of PD-L1 and PD-L2 is observed in erythroid cells during pregnancy. Moreover, the percentage of PD-L1+ or PD-L2+ erythroid cells is higher in the placenta than in the spleen of pregnant mice. However, fetal liver erythroid cells do not express PD-L1 or PD-L2. Blockade of PD-L1 restores immunoreactivity during pregnancy, which is expressed in increased production of interferon-gamma by T cells of pregnant mice [11]. Erythroid cells of premature newborns have a reduced expression of the TGFβ1 gene. Interestingly, premature infants have a higher content of erythroid cells compared to full-term ones [18]. Changes in the immunoregulatory potential of erythroid cells can probably contribute to increased immunoreactivity against the fetus, which leads to preterm birth.

Allogeneic pregnancy is accompanied by an immunological conflict; tolerance mechanisms make it possible for pregnancy to occur, preventing the mother’s body from responding to paternal antigens. In addition, the mechanisms of tolerance allow the fetus to develop and grow throughout the entire period of pregnancy, starting from the events of embryo implantation and up to the moment of initiation of labor. The suppressive phenotype of immune cells in the placenta is replaced by an inflammatory one at the end of gestation and initiates labor.

We suggest that placental erythroid cells play an important role in the induction and development of maternal tolerance and also take part in the change of the placental microenvironment from suppressive to inflammatory at the end of pregnancy. Syngeneic pregnancy was chosen in our study as a control group since it is practically not accompanied by an immunological conflict. 

In this work, we decided to evaluate the numbers of unsorted erythroid cells and secretomic profiles of CD71 magnetically enriched erythroid cells (CECs) in the mouse placenta at E12.5 (end of the second trimester) and E19.5 (end of the third trimester) in AP and SP.

## 2. Results

### 2.1. The Number of Erythroid Cells Decreases in Late-Term Pregnancy

We measured the percentage of erythroid cells in the placenta at E12.5 and E19.5 in both AP and SP (Figure 1) and found a significant 2.5-fold decrease in the percentage of erythroid cells in the placenta at E19.5 compared to E12.5, regardless of the type of pregnancy (AP or SP) (Figure 2).

### 2.2. Placental Erythroid Cells Are Mainly Represented by CD45^+^ Erythroid Cells

We then applied HSNE dimensionality reduction of erythroid cells’ flow cytometry data (FSC, SSC, CD44, CD45, CD71, and Ter-119) following *arcsinh*-transformation with automated co-factors (to automatically separate marker-expressing and marker non-expressing cells) and *fdaNorm*-normalization (to remove flow cytometry’s batch effects and normalize data). The results of the clustering reveal the presence of CD45^−^ and CD45^+^ erythroid cells (CD71^+^) and CD45^−^ and CD45^+^ reticulocytes (CD71^−^) (Figure 3).

Analysis of erythroid cells showed that CD45^+^ erythroid cells are predominant in the placenta and the presence of statistically significant changes in the structure of the erythron—we found that (a) there is a higher number of CD45^+^ erythroid cells and both CD45^−^ and CD45^+^ reticulocytes in AP E19.5 compared to E12.5 and (b) there is a lower number of CD45^+^ erythroid cells in SP E19.5 compared to E12.5 (Figure 4).

### 2.3. CECs from the Placenta at E12.5 in AP Have a Different Secretome Compared to Other Studied Terms and Types of Pregnancy

We studied CECs’ cytokine secretome of the placenta in AP or SP at E12.5 or E19.5. Secretion of CXCL1, CCL2, CCL3 and CCL4 was detected on every measured day of embryonic development and in every pregnancy type, while other cytokines’ secretion was inconsistent (Figure 5). 

CXCL1, CCL2, CCL3 and CCL4 chemokines were enriched in immune cells’ chemotaxis GO Biological Processes terms (Figure 6, Table 1).

Next, we analyzed differentially secreted cytokines in CECs. We found (a) significantly higher IFN-g secretion by placental CECs at E19.5 AP compared to E19.5 SP (Figure 7a,b) a total decrease in the secretion of all studied cytokines (especially of CCL11 and IL12p70) by CECs in the placenta by E19.5 AP compared with E12.5 AP (Figure 7b,c) significantly higher secretion of all studied cytokines (especially of CCL11 and IL12p70) by CD71-enriched placental erythroid cells at E12.5 AP compared to E12.5 SP (Figure 7c,d) an equal level of secretion of all studied cytokines by CECs in the placenta at E12.5 SP compared to E19.5 SP (Figure 7d).

## 3. Discussion

In this work, we investigated the composition of erythroid cells at various stages of differentiation and their secretion in the placenta of E12.5 or E19.5 mice with AP or SP.

We found that placental erythroid cells are mainly represented by CD45^+^ erythroid cells. According to the literature data, CD45^+^ erythroid cells have pronounced immunosuppressive properties [24]. We also found that with the course of pregnancy, the content of erythroid cells in the placenta decreases, both in AP and in SP. This can be explained by the need to remove the immune tolerance provided by the suppressive effect of erythroid cells and prepare for childbirth.

The secretome of CECs varied with both the day of intrauterine development and the type of pregnancy; CECs at E12.5 AP had a significantly higher level of secretion of all studied cytokines, especially for CCL11 and IL12p70. This can be explained by the following: (a) the very fact of allogeneic pregnancy and the presence of an immunological conflict attracts activated cells of the immune system to the placenta, which carry CD71 on their surface and contribute with their cytokines in the secretion of cells enriched in CD71 at E12.5 which disappear by E19.5; (b) erythroid cells produce a variety of cytokines that are differently directed towards inflammation under the influence of local placental factors, which are present only in AP and only at E12.5, which almost completely disappear by E19.5—secretion of only IFN-γ remains elevated at E19.5 with AP versus SP. Because IFN-γ is a primary pro-inflammatory cytokine, its residual presence in late gestation may help trigger the maternal immune response and induce labor.

For CCL11, it has been shown that it can regulate EVT migration, invasion and adhesion during uterine decidual spiral arteriole remodeling in the first trimester of human pregnancy [25]. It is possible that this chemokine is necessary for the remodeling of the spiral arterioles in AP, and its decrease to E19.5 compared to E12.5 is because the remodeling of the uterine spiral arterioles has already been completed and there is no need for it.

IL-12p70 alone does not affect erythropoiesis, but in combination with IL-4, it activates the differentiation of immature hematopoietic cells in the erythroid direction and promotes the growth of BFU-E [26]. For Homo sapiens, it is described that IL-4 suppresses IL-3-dependent erythroid colony formation by normal bone marrow cells, but in the presence of IL-3 and erythropoietin, IL-4 stimulates erythroid colony formation [27,28]. IL-4 is actively produced by placental Th2 cells since it is important for maintaining a state of immune silence in the uterus during pregnancy. In this case, the high production of IL12p70 by CD71-rich erythroid cells can be explained by the need to correct gestational anemia.

The only cytokines whose secretion was observed on all days of intrauterine development and in all types of pregnancy were the chemokines CXCL1, CCL2, CCL3 and CCL4, although their secretion was also significantly higher at E12.5 in AP. We hypothesize that these chemokines act as a bait for other immune system cells that have arrived in the placenta, trapping them in erythroid cells, which in turn induce functional anergy of such immune system cells with the help of PD-L1, arginase-1 [11] and TGF-β1 [16,29]. It has been described that, in addition to Tregs, tumor-associated macrophages and myeloid suppressor cells, erythroid cells are found in the tumor microenvironment. By producing IL-10 and TGF-β1, erythroid cells cause local immunosuppression, and erythroid cells are more powerful suppressors than MDSCs and Tregs [24]. Interestingly, we did not detect any IL-10 production by the E12.5 and E19.5 SP CECs.

It was previously described that the removal of erythroid cells led to fetal resorption [11]. It is possible that with a decrease in the relative number of erythroid cells, the ability of erythroid cells to lure immune system cells that enter the placenta with chemokines and trigger anergy in them also decreases. The secretion of the same chemokines (except for CCL4) by the rat myometrium was increased after delivery [30]. If the mouse myometrium also secretes these chemokines at the time of and after childbirth, then with the disappearance of the secretion of these chemokines by erythroid cells that have left the placenta, placental immune cells that migrated earlier to erythroid cells will migrate to the myometrium, which can lead to an increase in the immunological conflict (due to the loss of inhibitory action of erythroid cells) and, potentially, activation of labor activity.

## 4. Materials and Methods

### 4.1. Mice

To elucidate the role of erythroid cells in the process of gestation, allogeneic pregnancy ♀CBA × ♂C57Bl6 and syngeneic ♀CBA × ♂CBA pregnancy were modeled. One male and one female of the corresponding genetic line were put into a cage. Animals received food and drink as needed. The onset of pregnancy was recorded by the appearance of a vaginal plug in females. Euthanasia and organ retrieval were performed on days 12–13 post coitum (mid-term/E12.5) and on days 19–20 (late-term/E19.5). Euthanasia of pregnant mice was carried out using cervical dislocation.

We obtained mice from the vivarium of the Institute of Cytology and Genetics (Novosibirsk). Mice lived in conventional vivarium conditions with water and food access ad libitum, under the natural dark/light cycle. All experiments using mice were approved by the ethics committees of RIFCI. 

### 4.2. Cell Isolation

After euthanasia in pregnant mice under aseptic conditions, the abdominal cavity was opened, with the uterine horns being carefully cut; the fetuses were removed along with the placentas, and they were transferred to a sterile Petri dish, where the fetus was separated from the placenta. The isolated placentas were placed in sterile glass vials with cold RPMI-1640 medium containing antibiotics. Placentas obtained from one mouse were pooled. Isolation of placental cells was carried out by homogenizing whole placentas in a glass homogenizer with a pestle in 3–5 mL of PBS. Large tissue remnants were removed from the cell suspension with a pipette. After washing the cell suspension in 5–10 mL PBS we centrifuged placental cells in density gradient Ficoll–Urografin (ro = 1119 g/cm^3^) for 30 min at 322 RCF and washed them twice in PBS to remove RBCs and granulocytes. We separated cells after counting via magnetic separation.

### 4.3. Magnetic Separation

We performed magnetic separation of mononuclear splenic cells and marrow cells using anti-CD71-biotinylated antibodies (#113803, Biolegend, San Diego, CA, USA) and streptavidin-linked magnetic beads (#480015, Biolegend, Biolegend, San Diego, CA, USA) according to the manufacturer’s protocols (MojoSort™ Streptavidin Nanobeads Column Protocol—Positive Selection, https://www.biolegend.com/protocols/mojosort-streptavidin-nanobeads-column-protocol-positive-selection/4773/ accessed on 30 January 2023).

### 4.4. Viability Staining

We measured the viability of magnetically separated CD71^+^ erythroid cells (CECs) on a Countess 3 Automated Cell Counter (Thermo Fisher Scientific, Waltham, MA, USA) according to the manufacturer’s protocols using Trypan Blue. Trypan Blue staining showed > 95% viability for the sorted CECs. 

### 4.5. Erythroid Cells’ Flow Cytometry

We washed 0.5 × 10^6^ cells in PBS with the addition of 0.09% NaN_3_ and stained them with the antibodies according to the manufacturer’s protocols. We used Pacific Blue anti-mouse TER-119/Erythroid Cells Ab #112.5232, PE anti-mouse/human CD44 Ab #103024, FITC anti-human CD45 Ab #304006 and APC anti-mouse CD71 Ab #113820 (Biolegend, San Diego, CA, USA). We then washed the cells after 30 to 40 min of incubation in the dark with 0.5 mL PBS with the addition of 0.09% NaN_3_. We added 7-AAD to all samples right before cytometry. We conducted flow cytometry on an Attune NxT flow cytometer (Thermo Fisher Scientific, Waltham, MA, USA). 

### 4.6. Cell Culturing

We cultured the magnetically sorted cells in X-VIVO 10 serum-free medium with the addition of ×1 Insulin-Transferrin for 24 h at a concentration of 1 million per mL of the medium to support their viability and measure culture media cytokines afterward.

### 4.7. Erythroid Cells’ Conditioned Media Harvesting

We separated erythroid cells’ conditioned media from cells after 24 h of culturing. We performed the separation by centrifugation at 1500 rpm for 10 min, transferred the cell conditioned media into new 1.5 mL tubes with the addition of BSA up to the total concentration of 0.5% and froze the cells’ conditioned media at −80 °C until the cytokine quantification.

### 4.8. Cytokine Quantification in Culture Medium

We prepared 50 μL of each CECs conditioned media sample (n = 3–5) for a cytokine quantification with a Bio-Plex Pro mouse cytokine 23-Plex assay (#M60009RDPD, BioRad, Hercules, CA, USA) according to the manufacturer’s recommendations and analyzed them on the Bio-Plex 200 instrument (BioRad, Hercules, CA, USA).

### 4.9. Data Analysis: Flow Cytometry Data 

We manually gated populations of interest (Ter-119^+^ cells) in conventional gating software (for Attune NxT) and exported them as .fcs files. The .fcs files were then transformed to .csv files using a custom Python 3 Jupyter Notebook. The .csv files containing flow cytometry data were *arcsinh*-transformed, *fdaNorm*-normalized and exported as .fcs files with the R script published by Melsen et al. [31]. We then exported .fcs files into Cytosplore [32] and identified erythroid cell clusters using prior knowledge—CD71 expression disappears at the reticulocyte stage, and Ter-119 is present on all mouse erythroid cells. We then exported clusters’ cell count data into GraphPad Prism 9.4 app for macOS. We used either one-way ANOVA with Tukey correction for multiple testing (for the analysis of erythroid cells in bulk) or two-way ANOVA with Tukey correction for multiple testing for our statistical analyses (for intercluster analysis of erythroid cells). 

### 4.10. Data Analysis: Cytokine Secretion Data

We log_2_-transformed our Bio-Plex cytokine data using Pandas. We created the heatmap via Bioinfokit [33]. We performed the Gene Ontology enrichment analysis via GSEApy [34]. We performed multiple T-tests with FDR correction for differential cytokine production analyses in GraphPad Prism 9.4 app for macOS. We considered FC > 1.8 or FC < −1.8 and *q*-values < 0.05 statistically significant.

## 5. Conclusions

Placental erythroid cells are mainly represented by CD45^+^ immunosuppressive erythroid cells and secrete the chemokines CXCL1, CCL2, CCL3 and CCL4. It is known that these chemokines provide chemotaxis of many types of cells of the immune system, and therefore could potentially act as “bait” attracting immunocompetent T cells to the placenta. We assume that immune cells, having fallen into the “trap” of placental erythroid cells, may become anergic or tolerogenic, which ensures the development and maintenance of pregnancy. 

## Figures and Tables

**Figure 1 ijms-24-08130-f001:**
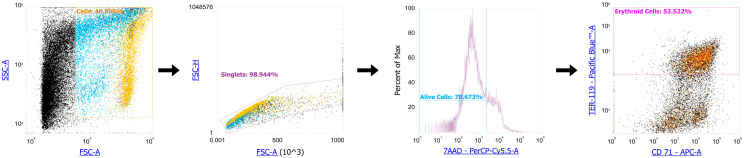
Erythroid cell gating strategy for further clustering-based analysis.

**Figure 2 ijms-24-08130-f002:**
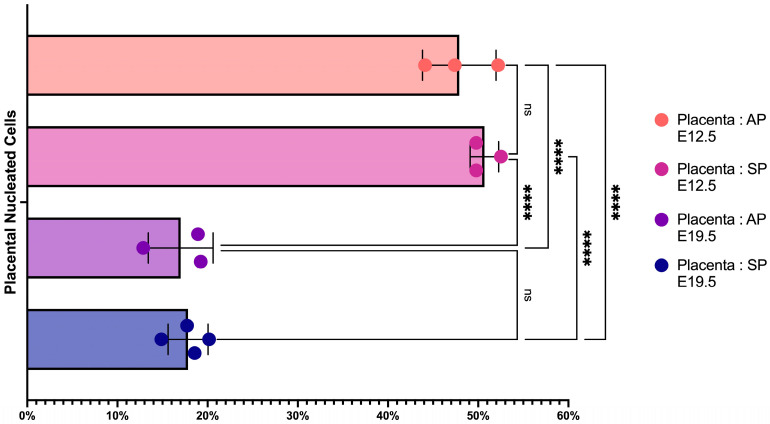
Percentage of erythroid cells in the placenta from either AP or SP at different time points of the total number of placental nuclear cells. E stands for the day of embryonic development. **** depict statistically significant differences (*q*-value < 0.001).

**Figure 3 ijms-24-08130-f003:**
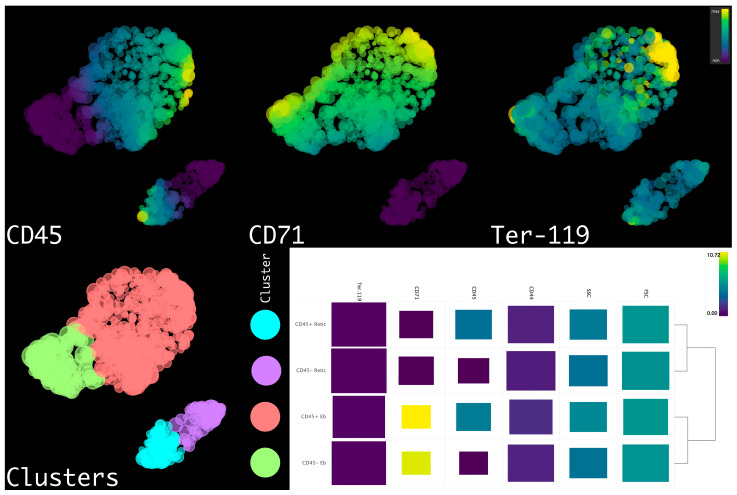
Integrated HSNE plot of erythroid cells from the erythroid cells in the placenta from either AP or SP at different time points with flow cytometry data; clusters are color-labeled in accord with the heatmap, and rectangle sizes are inversely proportional to marker variations. Eb stands for erythroblast, Retic stands for reticulocyte.

**Figure 4 ijms-24-08130-f004:**
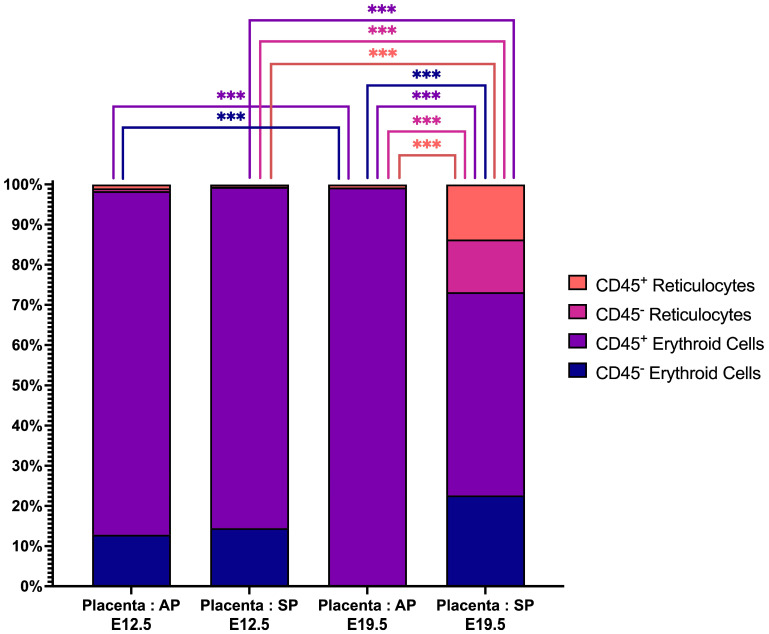
The percentages of erythroid cells at successive stages of differentiation in the placenta from either AP or SP at E12.5 or E19.5. E stands for the day of embryonic development. *** depict statistically significant differences (*q*-value < 0.005).

**Figure 5 ijms-24-08130-f005:**
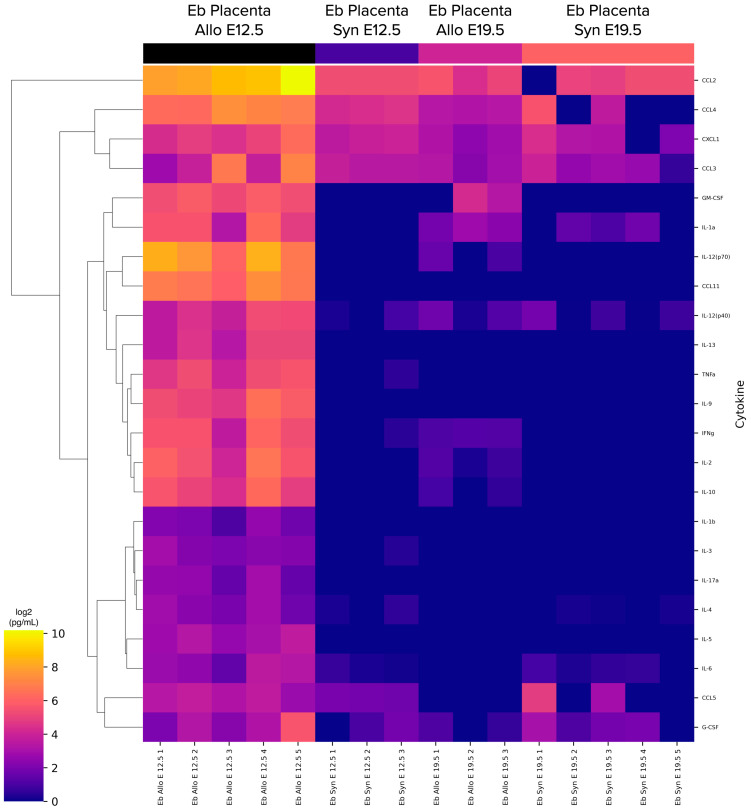
Heat map of the cytokines secreted by the CECs. Eb stands for CECs, Allo stands for AP placenta, Syn stands for SP placenta, and E stands for the day of embryonic development.

**Figure 6 ijms-24-08130-f006:**
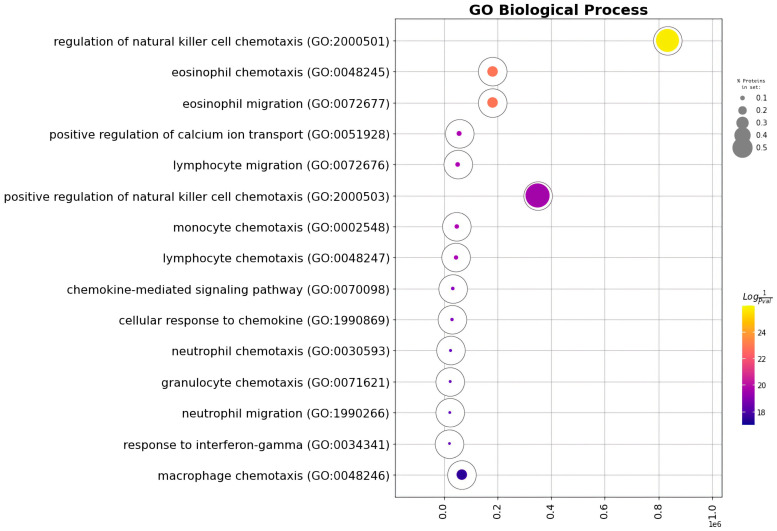
Ring plot of the Gene Ontology enrichment analysis of the cytokines most stably secreted by CECs.

**Figure 7 ijms-24-08130-f007:**
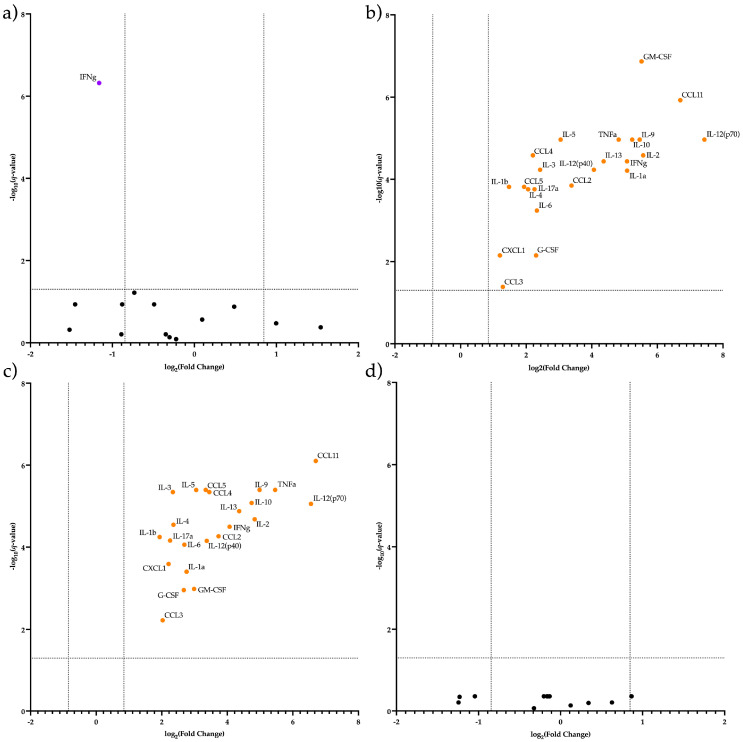
Differentially Secreted Cytokines’ Volcano plot of: (**a**) CECs from E19.5 SP Placenta vs. CECs from E19.5 AP Placenta (**b**) CECs from E12.5 AP Placenta vs. CECs from E19.5 AP Placenta, (**c**) CECs from E12.5 AP Placenta vs. CECs from E12.5 SP Placenta, (**d**) CECs from E12.5 AP Placenta vs. CECs from E12.5 SP Placenta.

**Table 1 ijms-24-08130-t001:** Gene Ontology enrichment analysis of the cytokines most stably secreted by CECs.

GO Term	Overlap	Q-Value	Score	Proteins
Chemokine-mediated signaling pathway	4/56	0.000000536	1,884,465	CCL4, CCL3, CCL2, CXCL1
Cellular response to chemokine	4/60	0.00000536	1,861,481	CCL4, CCL3, CCL2, CXCL1
Regulation of natural killer cell chemotaxis	3/7	0.00000536	344,521.8	CCL4, CCL3, CCL2
Neutrophil chemotaxis	4/70	0.00000536	1,810,210	CCL4, CCL3, CCL2, CXCL1
Granulocyte chemotaxis	4/73	0.00000536	1,796,267	CCL4, CCL3, CCL2, CXCL1
Neutrophil migration	4/77	0.000000555	1,778,550	CCL4, CCL3, CCL2, CXCL1
Eosinophil chemotaxis	3/16	0.00000344	93,174.6	CCL4, CCL3, CCL2
Eosinophil migration	3/16	0.00000344	93,174.6	CCL4, CCL3, CCL2
Inflammatory response	4/230	0.0000311	1,414,560	CCL4, CCL3, CCL2, CXCL1
Positive regulation of calcium ion transport	3/37	0.0000382	30,956.93	CCL4, CCL3, CCL2
Lymphocyte migration	3/40	0.0000441	28,054.03	CCL4, CCL3, CCL2
Monocyte chemotaxis	3/42	0.0000441	26,382.39	CCL4, CCL3, CCL2
Lymphocyte chemotaxis	3/44	0.0000005	24,884.33	CCL4, CCL3, CCL2
Response to interferon-gamma	3/80	0.000288	11,810.81	CCL4, CCL3, CCL2
Positive regulation of natural killer cell chemotaxis	2/5	0.000314	100,096.6	CCL4, CCL3
Response to interleukin-1	3/86	0.000314	10,795.79	CCL4, CCL3, CCL2

## Data Availability

The data that support the findings of this study are available from the corresponding author, S.V. Sennikov, upon an email request.

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
