# Peer review of "Murine Placental Erythroid Cells Are Mainly Represented by CD45+ Immunosuppressive Erythroid Cells and Secrete CXCL1, CCL2, CCL3 and CCL4 Chemokines"

_ijms, 2023, doi:10.3390/ijms24098130_

Round 1

Reviewer 1 Report

In this manuscript entitled " The secretomes of placental erythroid cells of mice in allogenic pregnancy differ from those in syngeneic pregnancy, the authors analyze the protein expression characteristics of placental erythrocytes in allogeneic and syngeneic pregnancies in mice. 

Although the data and its hypotheses are interesting, the reviewers are not convinced that the central claims of the manuscript are correct. Some of the key conclusions stated in the manuscript are not adequately supported by the data presented and are largely the result of speculation. Many of the objectives for each experiment are unclear, so the reviewers cannot be sure whether the results of each experiment support the authors' ideas and have been achieved. 

Comments: 

1. The title is plainly mediocre. The authors should give a specific title derived from the findings. 

2. The reviewer did not know why molecular markers such as FSCs and CD44 were examined in the flow cytometry experiments; in the Introduction chapter, please describe the analysis of proteins expressed by placenta-derived erythrocyte cells that are known so far. 

3. The authors should explain in detail what the experiment in Figure 1 aims to do. Please provide further explanation of the legend in Figure 1. 

4. The horizontal axis in Figures 2 and 4 is not clear. Is it the survival rate or the percentage of other items? Please provide further explanation of the legend in Figures 2 and 4. 

5. The reviewer does not know why you decided on the timing of the analysis of the percentage of erythrocyte cell viability (?). The authors should write down their reasons and thoughts on why they decided on that point (E12.5 and E19.5). 

6. It is unclear what the purpose of the analysis by flow cytometry was. The reviewer speculated that the results in Fig. 1 suggest a mixture of different erythroid cells. Therefore, is the aim to sort out specific erythrocytes involved in placental function? The authors should write their conclusions in Figure 1, followed by the purpose of the experiment in Figure 2, as appropriate. 

7. It is unclear how the differentiation status of erythrocyte cells was determined. The reviewer could not find any description of the method of differentiation state distinction. Please describe how to distinguish between differentiated states in the Materials and Methods chapter. 

8. The authors should also overview the differentiation process of erythrocytes and its significance in the placenta in the Introduction chapter. Next, it is necessary to write down the authors' objectives in Figure 4.  

9. Why is the proportion of early basophilic erythroid cells higher in AP and the proportion of polychromatophilic erythroid cells higher in SP? Is this an expected result, please discuss in the Discussion chapter. 

10. Why was CD71 used to isolate erythrocyte cells? Because CD71 is highly expressed in early basophilic erythroblasts? The authors should write carefully about this reason.

Spell checking should be reconsidered.

Author Response

Dear Reviewer, thank You for Your Review! 

We tried our best to fix every weak aspect of the manuscript. We also completely re-did our cluttering analysis, because we initially forgot to use CD45 for the clustering. New erythroid cell clustering is now easy to visually understand and revealed that most of the placental erythroid cells are in fact CD45 positive. This is important as such erythroid cells were shown to be especially immunosuppressive [https://www.frontiersin.org/articles/10.3389/fimmu.2022.830025/full, https://www.frontiersin.org/articles/10.3389/fimmu.2020.597433/full]. We completely re-did Figures 3 and 4, we also fixed Figures 2 and 5. We also fixed a lot of spelling mistakes.

1. The title is plainly mediocre. The authors should give a specific title derived from the findings. 

We changed the title to include our new findings on CD45:

«Murine placental erythroid cells are mainly represented by CD45+ immunosuppressive erythroid cells and secrete CXCL1, CCL2, CCL3 and CCL4 chemokines.»

2. The reviewer did not know why molecular markers such as FSCs and CD44 were examined in the flow cytometry experiments; in the Introduction chapter, please describe the analysis of proteins expressed by placenta-derived erythrocyte cells that are known so far. 

Erythroid cells do not have tissue-specific markers, as far as the literature shows. We used markers that were used in previous works. As we re-did the clustering and CD45 took the spotlight, FCS and CD44 became non-important to the results.

3. The authors should explain in detail what the experiment in Figure 1 aims to do. Please provide further explanation of the legend in Figure 1. 

The steps described in Figure 1 show the data pre-processing pipeline for further clustering in Cytosplore. It is required to manually gate the population of interest. We expanded Figure 1’s legend to be: «Erythroid cells’ gating strategy for the further clustering-based analysis.»

4. The horizontal axis in Figures 2 and 4 is not clear. Is it the survival rate or the percentage of other items? Please provide further explanation of the legend in Figures 2 and 4. 

Figure 2 shows the percentage that placental erythroid cells make from the total nucleated placental cells.

Figure 4 was redone and now shows a stacked bar plot that shows percentages that different clusters make from the total placental erythroid cells.

5. The reviewer does not know why you decided on the timing of the analysis of the percentage of erythrocyte cell viability (?). The authors should write down their reasons and thoughts on why they decided on that point (E12.5 and E19.5). 

Erythroid cell viability was high at every measured time point and type of pregnancy. These days (12.5 and 19.5) of embryonic development were chosen as the final days of the second and third trimesters respectively of murine pregnancy which usually lasts for 19 to 21 days. We mentioned this in the introduction: «In this work, we decided to evaluate the numbers of unsorted erythroid cells and secretomic profiles of CD71-magnetically-enriched erythroid cells (CECs) in the mouse placenta at E12.5 (end of the second trimester) and E19.5 (end of the third trimester) in AP and SP.»

6. It is unclear what the purpose of the analysis by flow cytometry was. The reviewer speculated that the results in Fig. 1 suggest a mixture of different erythroid cells. Therefore, is the aim to sort out specific erythrocytes involved in placental function? The authors should write their conclusions in Figure 1, followed by the purpose of the experiment in Figure 2, as appropriate. 

We measured cellular abundance at different days of embryonic development, higher numbers of erythroid cells would suggest a higher local immunosuppression. We were indeed able to deduce placental-specific erythroid cells in Figure 3 - CD45+ erythroid cells that make nearly 90% of the total erythroid cells. In comparison - murine bone marrow CD45+ erythroid cells only make 2-5% of the total erythroid cells.

7. It is unclear how the differentiation status of erythrocyte cells was determined. The reviewer could not find any description of the method of differentiation state distinction. Please describe how to distinguish between differentiated states in the Materials and Methods chapter. 

We removed that part as clustering-based analysis revealed a great number of CD45+ erythroid cells.

8 and 9. The authors should also overview the differentiation process of erythrocytes and their significance in the placenta in the Introduction chapter. Next, it is necessary to write down the authors' objectives in Figure 4. Why are the proportion of early basophilic erythroid cells higher in AP and the proportion of polychromatophilic erythroid cells higher in SP? Is this an expected result, please discuss it in the Discussion chapter. 

We were not able to find any data on the significance of the differentiation process of erythroid cells in the placenta. As we were not sure about our differentiation clustering, we removed the old Figure 4.

10. Why was CD71 used to isolate erythrocyte cells? Because CD71 is highly expressed in early basophilic erythroblasts? The authors should write carefully about this reason. 

Enrichment of erythroid cells by positive magnetic separation is the most common method in the field, first article dates back to 2013 [https://pubmed.ncbi.nlm.nih.gov/24196717/]. The authors of this paper claimed the term CD71+ erythroid cells (CECs). The reasoning behind the method is that CD71 is expressed on all erythroid cells from the burst forming unit (erythroid) to the orthochromatophilic erythroblast state [https://academic.oup.com/ajcp/article/134/3/429/1766450]. We added this information to the Introduction section: «However, some researchers tend to use the other marker for the erythroid cells’ studies - CD71. Such cells are named CD71+ erythroid cells (CECs) [8].»

Reviewer 2 Report

Section

Comments and recommendation

General comment

The manuscript entitled: The secretomes of plancental erythroid cells of mice allogeneic pregnancy differ from those in syngeneic pregnancy; which deals with an advanced research topic using cytokine secretion approach. Indeed, this research topic has a wide range of interest and downstream applications and within the scope of International Journal of Molecular Sciences. Moreover, the authors have used the standards of manuscript writing according to Journal format.

Abstract

1-The abstract is well structured and described well all contents of this investigation. The authors have highlighted the objectives clearly to be well attached with the purpose of this investigation.

2- The conclusion summarized well the key findings of this investigation.

Introduction

1-The introduction described well the key problem that this manuscript is dealing with and up-to-date literature connected to the topic.

2-Moreover, the literature justifies and reinforces the desire to conduct such research work and highlights the gap of research.

Materials and methods

1-The experiment design and the work flow of this investigation is well constructed and is fitting with the objectives.

 2- The research methodologies which have been applied in this investigation are relevant fitting well to the design and objectives. 3-The bioinformatics tools and statistical programs used for secretome analysis are suitable for these types of data. The ethical approval was taken before performing this study.

Results

The results are well described in the manuscript.

Discussion

1-The discussion section is well written and results are well discussed in the discussion section. The authors have used literature that is attached to the topic.

2- The discussion of the results addresses the topic of this study in understandable manner and highlights the core findings of the investigations.

3-The authors highlights the significance of secretome composition in relation to type of pregnancy (allogeneic vs. syngeneic).

Conclusion

The conclusion summarized well the core findings of this investigation.

Bibliography/References

The list of references is well organized according to journal standard style.

Recommendation and final comment

I recommend acceptance of manuscript entitled: The secretomes of plancental erythroid cells of mice allogeneic pregnancy differ from those in syngeneic pregnancy.  This decision was made according to the significance value of the manuscript findings in addition to its originality. 

Author Response

Dear Reviewer, thank You for Your Review!

Round 2

Reviewer 1 Report

The reviewers' concerns were completely removed.